# Deep Deterministic Policy Gradient with Reward Function Based on Fuzzy Logic for Robotic Peg-in-Hole Assembly Tasks

**Ziyue Wang** [1], **Fengming Li** [2,3,*,†], **Yu Men** [3], **Tianyu Fu** [3], **Xuting Yang** [3] **and Rui Song** [3]

1   College of Science, Guilin University of Technology, Guilin 541006, China; wziyins27@163.com
2   School of Information and Electrical Engineering, Shandong Jianzhu University, Jinan 250101, China
3   School of Control Science and Engineering, Shandong University, Jinan 250061, China;
    202114785@mail.sdu.edu.cn (Y.M.); futy0807@gmail.com (T.F.); 201914426@mail.sdu.edu.cn (X.Y.);
    rsong@sdu.edu.cn (R.S.)
*   Correspondence: lifengming@sucro.org or lifengming21@sdjzu.edu.cn; Tel.: +86-186-6016-8885
†   Current address: School of Control Science and Engineering, Shandong University, Jingshi Road 17923,
    Jinan 250061, China.

**Abstract:** Robot automatic assembly of weak stiffness parts is difficult due to potential deformation during assembly. The robot manipulation cannot adapt to the dynamic contact changes during the assembly process. A robot assembly skill learning system is designed by combining the compliance control and deep reinforcement, which could acquire a better robot assembly strategy. In this paper, a robot assembly strategy learning method based on variable impedance control is proposed to solve the robot assembly contact tasks. During the assembly process, the quality evaluation is designed based on fuzzy logic, and the impedance parameters in the assembly process are studied with a deep deterministic policy gradient. Finally, the effectiveness of the method is verified using the KUKA iiwa robot in the weak stiffness peg-in-hole assembly. Experimental results show that the robot obtains the robot assembly strategy with variable compliant in the process of weak stiffness peg-in-hole assembly. Compared with the previous methods, the assembly success rate of the proposed method reaches 100%.

**Keywords:** robot assembly; deep reinforcement learning; fuzzy reward; compliant control

## 1. Introduction

The robot operating contact environment is changeable and unpredictable. It is a challenge that the robot could quickly perform new tasks and precisely control the contact force in different environments. High-precision assembly is a typical contact operation [1,2], and the assembly process needs to overcome the environmental model and controller errors. The peg-in-hole assembly process is usually divided into the search phase and the insertion phase [3], which is visual and tactile. In the insertion phase, the center axis of the peg-in-hole inserts into the bottom. When the axis deviation or force/torque is not appropriate, it can cause card resistance or wedge tightening. Due to the deformation error, friction and robot positioning error between assembly objects, it is difficult to establish an accurate physical model and find the optimal assembly strategy according to the model analysis.

Robot assembly control strategies could be designed with forces and torques in the robot assembly based on mathematical models. Compared to the position feedback controller with high gain, impedance control ensures that the robot and environment are fully controllable. A natural mass-damping-spring relationship is maintained between the contact force and the position offset, and its force control characteristics depend on inertia, stiffness, and damping parameters [4]. The traditional method of adjusting parameters manually adjusts the control parameters according to the characteristics of the task. For the assembly of such complex tasks, it is difficult to set the impedance control method of fixed parameters to achieve the target task. If the parameters of impedance control could be

adjusted to changes in the assembly tasks, the control performance is improved. The safety of the robot could also be guaranteed in the process of adjusting the impedance parameters.

An important feature of reinforcement learning [5,6] is that optimal performance could be achieved by designing incentive function guidance without understanding the robot and environmental system models. Therefore, this paper mainly studies how robots could react to contact forces by changing the stiffness motion in the robot peg-in-hole insertion task. The robot obtains the optimal control strategy in the assembly process through dynamic adjustment of the impedance parameters. Aiming at the soft control problem in the robot contact task, this paper combines the quality evaluation of fuzzy reward and the deep deterministic strategy gradient to achieve variable impedance. The robot could adapt to the different contact force changes in the assembly process.

The main contributions are as follows:

(1) The reward function with fuzzy logic based on assembly quality evaluation was designed in the actual industrial environment to improve the system learning performance;
(2) The assembly process model was constructed and the robot completed the stiffness workpieces assembly;
(3) The robot combined with the variable impedance in the course of operation to avoid the damage of the workpiece during the learning process;
(4) A framework of learning robot assembly skills is proposed, and it is verified in the peg-in-hole assembly of weak stiffness workpieces.
(5) The proposed method in this paper combines deep reinforcement learning with robotics technology, which provides theoretical support for the improvement of complex manipulation skills of a new generation of robots. It also provides a new idea for the application of artificial intelligence algorithms in the industrial field.

The remainder of this paper is organized as follows. Section 2 introduces the related work. The assembly system and formulation of the problem to be solved are in Section 3. Section 4 contains the description of the proposed method. Experiments were performed to validate the proposed method, and the results are presented and discussed in Section 5. Finally, in Section 6, we summarize the results of the current work and discuss future directions.

## 2. Related Work

The robot assembly strategy is divided into traditional control strategy and learning-based assembly control strategy. The traditional control strategy mainly depends on position-based impedance control, which produces submissive characteristics to the contact environment via assembling objects. That could effectively reduce jam or clam in order to complete the assembly work. The learning-based assembly control strategy is mainly to obtain the optimal assembly strategy using the learning algorithm, including the parameter optimization of impedance control model.

The traditional assembly strategy is that the robot uses the control algorithm to realize the assembly using the feedback information of the sensor [7]. The force control strategy is the main method of peg-in-hole assembly [8]. By analyzing the contraction state and contact force during assembly, it would reduce the contact force and torque to speed up the assembly process. Chan uses impedance control with combining force error and motion error to control SCARA robots to complete the PCB assembly [9]. Aljairah proposes a gradient-based control strategy to continuously update the robot for the smooth control of the assembly process [10]. Xu and Inouel et al. divide peg-in-hole assembly into search and jack phases [11]. Lin Junjian solves the problem of the zero value change of the force sensor through the gravity compensation system of the force sensor [12]. In order to achieve good assembly effect, the traditional control strategy requires a higher contact environment. The contact state is generally complex and dynamic [13], and all the information must be obtained by means of the accurate environmental contact model. In the actual industrial assembly scene, noise interference, environmental complexity and change could not be

achieved. The application based on the learning algorithm ensures the automatic operation of assembly work.

At present, the model-less intensive learning method is widely used in the manipulation task with rich contact characteristics [14,15]. Vijaykumar et al. [16] put forward a value function-based method, using the greedy strategy to select discrete assembly action with long short-term memory network (LSTM). The circular neural network estimation Q value function was used to achieve peg-in-hole assembly beyond the resolution of the robot [1]. The robot action based on the enhanced learning method of value function is discrete and low dimensional. M. Nuttin et al. [17] put forward two components, the actor and the critic, to derive the robot assembly strategy and evaluate the action performed. Jing et al. [18] propose a model-driven DDPG method, combined with the feedback exploration strategy, for multi-axis hole assembly. Guide policy search [15] combines track optimization models with neural network strategies to learn a wealth of operational skills. Luo et al. [2] propose the MDGPS (mirror descent guided policy search) method, which does not rely on joint torque and uses force–torque signals from wrist sensors to complete the assembly of deformable and rigid objects.

Stiffness control could be obtained via analyzing models. Inoue et al. [1] propose robotic skill acquisition methods that train push neural networks through intensive learning. However, the resulting high stiffness discrete action constrains the smoothness and safety of the assembly process. Erickson D. et al. [19] obtain the contact force error through force sensors and estimated environmental damping stiffness models online. The environmental models are combined with impedance control to achieve the expected force impedance control parameter. Under the indirect excitation of the observation signal, it is difficult to obtain accurate environmental damping parameters. Sun Xiao et al. [20] propose a method of impedance control parameter regulation based on fuzzy adaptive, and verify the robot's regulation of end contact force on the PUMA560 robot.

In [21], the time-varying stiffness can be reproduced by properly controlling the energy exchanged during the movement. It could ensure the stability of the robot if the robot moves freely or interacts with the environment. Ficiello et al. [22] combine Cartesian impedance modulation with redundant resolution to improve the performance of human–computer physical interactions. It is demonstrated that a variable impedance with an appropriate modulation strategy for parameter adjustment is superior to a constant impedance. Howard et al. [23] modulate the impedance modulation strategy by imitating human behavior. According to the learned human-like impedance strategy, Chao et al. [24] modulate the impedance curve online to enhance the flexibility and adaptability of the system.

The position-based impedance control could be used without accurate mathematical modeling of robot dynamics. Therefore, the position-based impedance control is also used in this paper to ensure the compliance of peg-in-hole assembly process.

## 3. Problem Formulation

In the peg-in-hole insertion phase, there are two states of jamming and wedging. If the contact force is not appropriate, it is easy to cause damage to the assembly object. Impedance control [25] ensures that the robot's contact with the operating environment is controlled, and the end contact force $f$ and displacement $\Delta x$ remain:

$$f = M\Delta\ddot{x} + D\Delta\dot{x} + K\Delta x \tag{1}$$

where $\Delta x$ represents the displacement of the actual position $x$ relative to the reference position $x_v$. $M$, $D$, and $K$ represent the inertia, damping, and stiffness. In the process of high-precision assembly, the robot's degree of freedom is constrained, the movement is slow and smooth, the robot assembly is regarded as a quasi-static process, the inertia and damping in the impedance control are ignored [26], and the stiffness control is used instead of the complete impedance control, that is, $f = K\Delta x$. Based on the stiffness control of intensive learning, the optimal strategy is found by trial and error, which maps the robot assembly state $S$ to the robot's action. In continuous motion space, the historical

information of assembly process also has an influence on the choice of strategy, so the state $S$ in this paper uses trajectory deviation, contact force deviation, assembly process representation, and the action of the robot arm includes joint angle and stiffness.

$$S = [F, M, \theta, \tau] \tag{2}$$

## 4. Method

Assembly strategy learning is actually the parameter updating and optimization, which requires a stable and efficient learning algorithm. This paper uses the deep deterministic strategy gradient (DDPG) algorithm to realize the learning of stiffness strategy and trajectory strategy.

With the Actor–Critic framework, the Actor gets a mapping from state $S$ to action $a(t)$, and the Critic uses the state action value function $Q(s, a \,|\theta^Q)$ to quantify the strategy calculated. Here, the Actor and Critic are represented by neural networks, with parameters $\theta^\mu$ and $\theta^Q$, respectively.

### 4.1. Assembly Policy Learning

The assembly process is divided into non-contact stage and contact stage. The initial position is often random during the assembly process. In the assembly contact state, there is a corresponding relationship between the robot action and the current state. Then the guidance strategy of the subsequent stage could be carried out in turn, which could complete the assembly action smoothly to protect the workpiece from damage. As shown in Figure 1, the mapping between the contact state and robot action is learned with the framework.

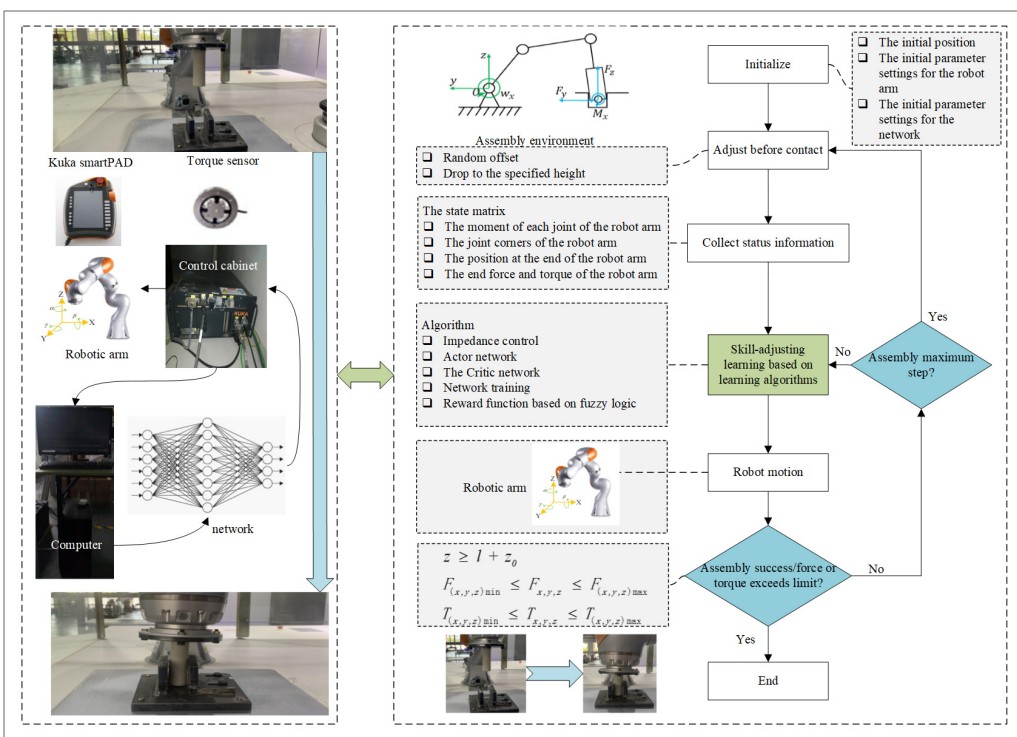

**Figure 1.** The framework of the proposed method.

The Actor shows the same network structure as shown in Figure 2. The hidden layer has three layers: the first layer contains 300 neurons, the second layer and the third layer each contains 200 neurons, the hidden layer and the output layer between the tanh activation function; the output layer gives an adjustment action, the output dynamic as the joint angle of the 7 axes and the stiffness of the three directions at the end of the robot arm

$$a = (\theta_1, \theta_2, \theta_3, \theta_4, \theta_5, \theta_6, \theta_7, k_x, k_y, k_z) \tag{3}$$

The logical relationships between layers are as follows:

$$layer_1 = ReLU(w_1 \times state + b_1) \tag{4}$$

$$layer_2 = softsign(w_2 \times layer_1 + b_2) \tag{5}$$

$$output = softsign(w_3 \times layer_2 + b_3) \tag{6}$$

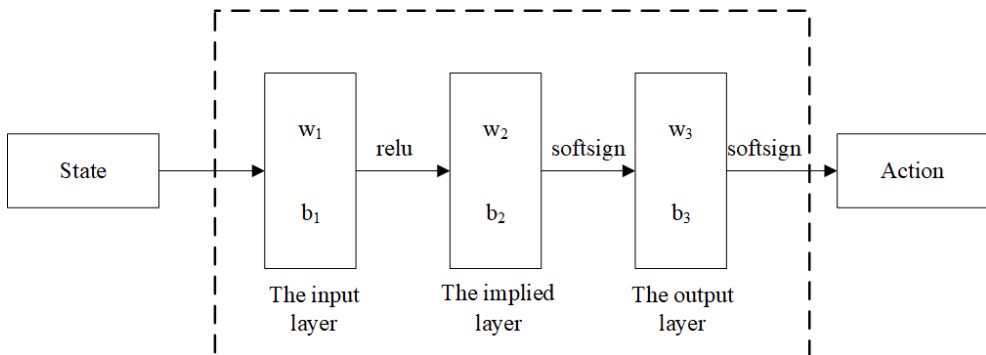

**Figure 2.** Actor network structure.

The Critic network structure is shown in Figure 3. According to the current contact state, the assembly action in the Actor network obtains the *Q* value, evaluates the assembly strategy, and then guides the actor network to make strategic adjustments. The input layer obtains the contact state information and the assembly action given by the Actor network, passes through the *ReLU* function into the implied layer, and then passes through the identity function into the output layer.

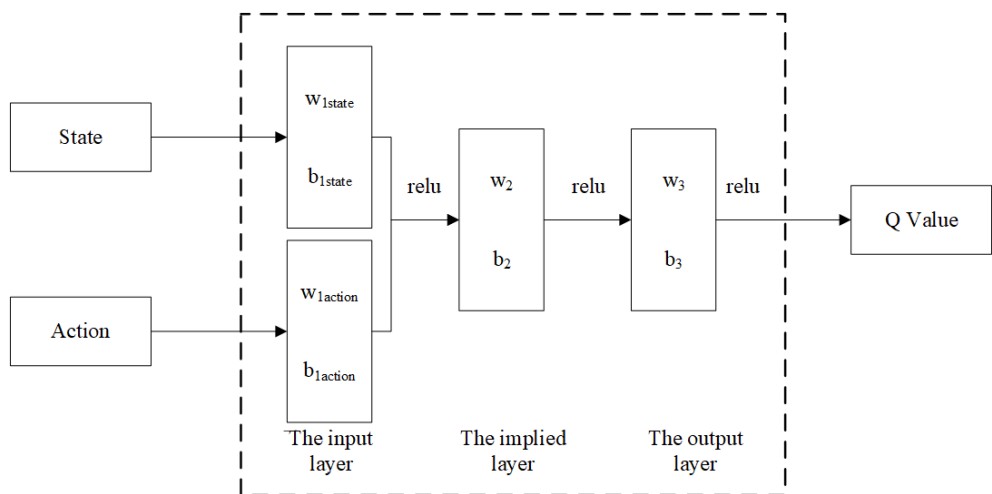

**Figure 3.** Critic network structure.

The logical relationships between layers are

$$layer_{1s} = ReLU(w_{1s} \times s + b_1) \tag{7}$$

$$layer_{1a} = ReLU(w_{1a} \times s + b_{1a}) \tag{8}$$

$$layer_2 = ReLU(w_{2s} \times layer_{1s} + w_{2a} \times layer_{1a} + b_2) \tag{9}$$

$$output = ReLU(w_3 \times layer_2 + b_3) \tag{10}$$

### 4.2. Reward Function

For peg-in-hole assembly, the assembly quality evaluation is one of the most important factors affecting the success rate. The fuzzy logic system could handle multiple parameters, so it could evaluate the quality of the assembly system very well. Therefore, this paper sets up a quality evaluation with fuzzy logic.

In this paper, four typical parameters are used as the parameters of fuzzy quality evaluation, i.e., contact force $F_Y$ of t-moment y axis, contact force $F_Z$ of z-axis, assembly depth $Z$ of t-moment shaft and assembly action amount $D_Z$:

$$F_Y = |F_y - f_y| \tag{11}$$

$$F_Z = |F_z - f_z| \tag{12}$$

$$Z = s_Z \tag{13}$$

$$D_Z = d - s_Z \tag{14}$$

where $F_z$ is the force of the *z* direction, $f_z$ is the initial force of the *z* direction, $F_y$ is the force of the *y* direction, $f_y$ is the initial force of the *y* direction, $s_Z$ is the $Z$ axis coordinates in the current state of the robot arm, and $d$ is the $Z$ axis coordinate value at the bottom of the hole under the same coordinate system as the robot arm.

If you blur these four parameters, 625 fuzzy rules are needed if you use only one layer of fuzzy logic system. In order to simplify the design of the reward system, the double-layer fuzzy logic structure of Figure 4 is adopted, the first layer has two fuzzy logic systems, the $F_y - F_z$ fuzzy logic system takes the contact force $F_y$ and z-axis contact force $F_z$ as input, and the $Z - D_Z$ fuzzy logic system takes the assembly depth $Z$ and assembly action $D_Z$ of the t-moment axis as the input. The output of the two systems serves as input to the second layer of fuzzy logic systems, and finally the value of the reward required is output by the second layer of fuzzy logic systems. Thus 75 fuzzy rules need to be made, greatly reducing the difficulty of fuzzy rules.

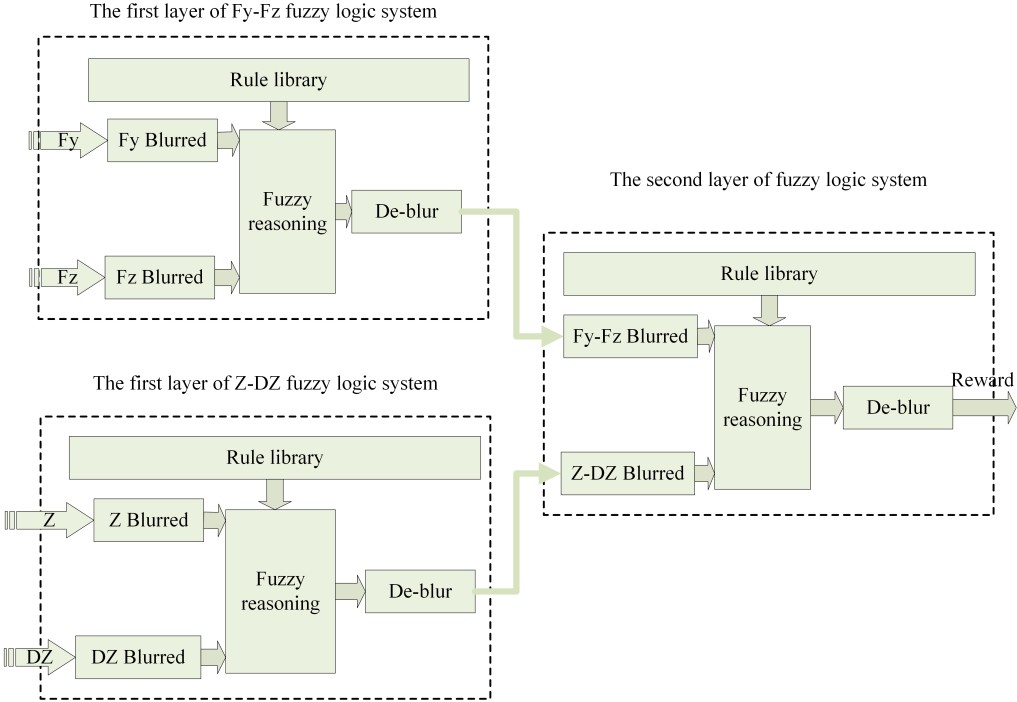

**Figure 4.** The assembly quality evaluation with fuzzy logic.

For the above parameter input system, using triangular membership function for fuzzy processing, each parameter is blurred into 5 fuzzy values: VG, G, M, B, and VB,

respectively, refer to very good, good, medium, bad, and very bad. Each fuzzy value has a membership range of $(0, 1)$.

$$
f(x) = \begin{cases} 0 & x \le a \\ \dfrac{x-a}{b-a} & a \le x \le b \\ \dfrac{c-x}{c-b} & b \le x \le c \\ 0 & x \ge c \end{cases}
\tag{15}
$$

where $a, b, c$ represent the parameter value in the triangular membership function, $a, c$ determines the width of the membership function, and $b$ determines the location of the membership function.

After blurring the parameters, fuzzy reasoning is based on the established rule library, and the rule library as shown in Table 1 is established according to the experience in peg-in-hole assembly. Additionally, use the AND operation for fuzzy reasoning.

$$
R_i(x) = min(\mu_{A(i)}(x), \mu_{B(i)}(x))
\tag{16}
$$

where $A, B$ are fuzzy collections. The AND operation takes the minimum value of the membership of both and determines the membership of the fuzzy value of the output according to the rule library.

**Table 1.** Rule library.

|  | **VG** | **G** | **M** | **B** | **VB** |
|---|---|---|---|---|---|
| VG | $R_1$ | $R_6$ | $R_{11}$ | $R_{16}$ | $R_{21}$ |
| G | $R_2$ | $R_7$ | $R_{12}$ | $R_{17}$ | $R_{22}$ |
| M | $R_3$ | $R_8$ | $R_{13}$ | $R_{18}$ | $R_{23}$ |
| B | $R_4$ | $R_9$ | $R_{14}$ | $R_{19}$ | $R_{24}$ |
| VB | $R_5$ | $R_{10}$ | $R_{15}$ | $R_{20}$ | $R_{25}$ |

Finally, the resulting fuzzy value is clearly processed, that is, the last part of the fuzzy logic system: de-fuzzing. Since there are many fuzzy values obtained after fuzzy reasoning, which cannot be used, the data need to be processed by the de-fuzzing method, and finally the clear value that meets our requirements is obtained.

Discrete fuzzy values are used in this paper. Use the center of gravity method to defuse

$$
U^* = \frac{\sum_i (C^*{}_i w_i)}{\sum_i (C^*{}_i)}
\tag{17}
$$

where $U^*$ refers to the clear output after de-blurring, $U^* \sim (-1, 0)$ refers to the fuzzy collection of fuzzy reasoning, and $w_i$ refers to the weight of each membership.

### 4.3. Network Training

During network training, the Actor and Critic's target networks are recorded as $\mu'(s|\theta^{\mu'})$ and $Q'(s, a|\theta^{Q'})$, which are used to calculate the target values. These target networks have to be updated with parameters

$$
\theta^{\mu'} = \tau\theta^{\mu} + (1-\tau)\theta^{\mu'}
\tag{18}
$$

$$
\theta^{Q'} = \tau\theta^{Q} + (1-\tau)\theta^{Q'}
\tag{19}
$$

$\theta^{\mu'}$ and $\theta^{Q'}$ are parameters of the target network, and $\tau = 0.0001$ is the progressive update rate for the target network.

During algorithm training, the Critic network optimizes parameter $\theta^Q$ by minimizing the loss function, Loss.

$$Loss = \frac{1}{N} \sum_i (y_i - Q(s_i, a_i | \theta^Q))^2 \tag{20}$$

Thereinto, $y_i = r_i + \gamma Q'(s_{i+1}, \mu'(s_{i+1}|\theta^{\mu'})|\theta^{Q'})$ $\gamma$ is a discount factor that balances the current and long-term penalties.

State action value function $Q$ updates Actor on the $\theta^\mu$ gradient:

$$
\begin{aligned}
& E[\nabla_{\theta^\mu} Q(s, a | \theta^Q)|_{s=s_i, a=\mu(s_i)}] \\
& = \frac{1}{N} \sum_i \nabla_a Q(s, a | \theta^Q)|_{s=s_i, a=\mu(s_i)} \nabla_{\theta^\mu} \mu(s | \theta^\mu)|_{s_i}
\end{aligned}
\tag{21}
$$

### 4.4. Algorithm Pseudocode

The Algorithm 1 pseudocode used in this article is as follows:

---

**Algorithm 1** Fuzzy Rewards—DDPG Algorithm

---

1: Initialize Actor–Critic's network parameters $\theta^Q$ and $\theta^\mu$.
2: Assign network parameters to the target network $\theta^{Q'} \leftarrow \theta^Q$, $\theta^{\mu'} \leftarrow \theta^\mu$.
3: Initialize the experience pool $R$.
4: **for** episode = (1, 1000) **do**
5:     Go back to the initial point $s_1$;
6:     **for** step = (1, 30) **do**
7:         Select Action $a_i = \mu(s_i | \theta^\mu)$ from the actor network and send it to the robot
8:         After processing
9:         The Fuzzy Rewards system calculates the reward value based on $F_y, F_z, \Delta_z, d_z$
10:         The program executes $a_i$ and returns the bonus value $r_i$ and the new status
11:         $s_{i+1}$
12:         The state transition process: $(s_i, a_i, r_i, s_{i+1})$ is stored in the experience pool $R$
13:         As a dataset for the training network
14:         Randomly sample N $(s_i, a_i, r_i, s_{i+1})$ data from Experience Pool $R$ as a mini-
15:         Batch training data for the policy network and Q network
16:         Set up $y_i = r_i + \gamma Q'(s_{i+1}, \mu'(s_{i+1}|\theta^{\mu'})|\theta^{Q'})$
17:         Define the loss function: $Loss = \frac{1}{N} \sum_i (y_i - Q(s_i, a_i | \theta^Q))^2$, update the Critic
18:         Network with the loss function
19:         Policy gradient for computing policy networks:
20:         $\nabla_{\theta^\mu} J = \frac{1}{N} \sum_i \nabla_a Q(s, a | \theta^Q)|_{s=s_i, a=\mu(s_i)} \nabla_{\theta^\mu} \mu(s | \theta^\mu)|_{s_i}$
21:         Update the target network parameters:
22:         $\theta^{\mu'} = \tau \theta^\mu + (1 - \tau)\theta^{\mu'}$
23:         $\theta^{Q'} = \tau \theta^Q + (1 - \tau)\theta^{Q'}$
24:     **end for**
25: **end for**

---

## 5. Experiments

### 5.1. Platform Building

The experimental platform of the KUKA LWR iiwa robot for information collection structure and action execution organization of assembly process is set up. The server is used to process the acquired assembly status data and complete the model learning to trigger the assembly control strategy, driving the robot to complete the flexible assembly task. Taking into account the precision of repeated positioning of the assembly robot, the peg-in-hole gap is set to 0.2 mm, stiffness (Yang's modulus) is simulated in this experiment, consisting of an axis fixed on the robot body and the assembly table to be assembled workpieces, for the completion of assembly tasks. Specific parameters are shown in Table 2. The shaft

of the assembly model consists of a steel shaft with a $E_{stl} = 2.1 \times 10^{11}$ Pa modulus of Yang's and a plastic sleeve with a Yang's modulus of $E_{stl} = 2.8 \times 10^6$ Pa, which produces less contact force than a pure stiffness shaft with the same displacement deviation.

**Table 2.** Assembly workpiece parameters.

| Diameter | Depth | Minimum Clearance | Center Deviation | Axis Angle |
|----------|-------|-------------------|------------------|------------|
| 24 mm | 30 mm | 0.5 mm | $(-5\text{ mm}, 5\text{ mm})$ | $(-0.05°, 0.05°)$ |

In this paper, the communication process is established between the assembly robot and the server through TCP/IP. As shown in Figure 5. The data transmission and command distribution are in the form of Socket. The iiwa robot communicates with the controller with Socket and receives feedback on the assembly status. The neural network is trained with the server. The initial parameter settings for assembly experiment are shown in Table 3.

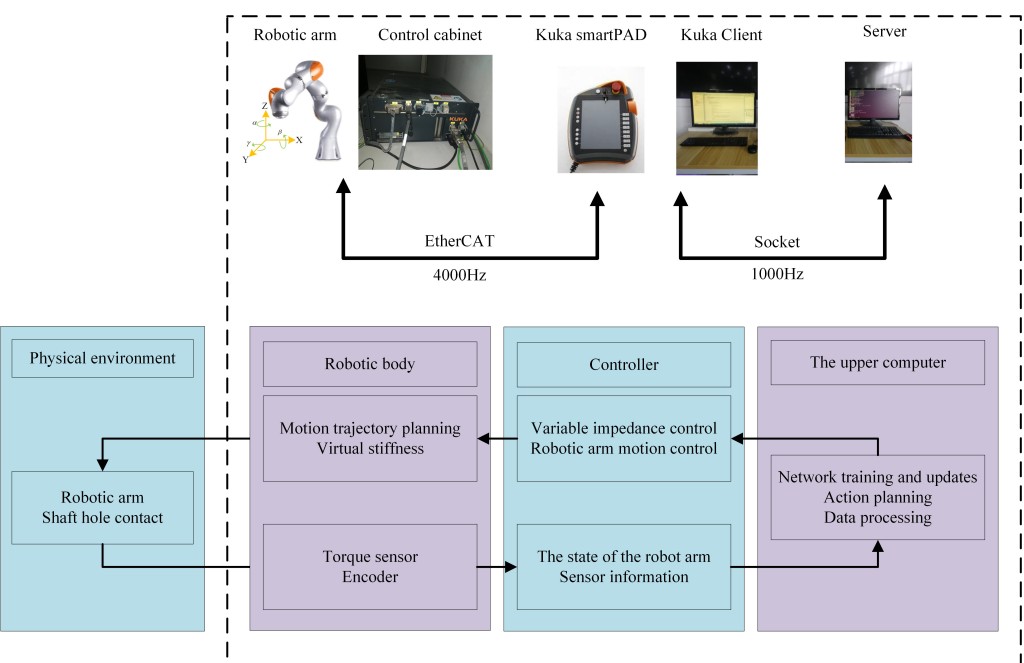

**Figure 5.** Communication structure of assembly system.

**Table 3.** Initial setting of each assembly experiment.

| Parameter | Symbol | Value |
|-----------|--------|-------|
| Contact force threshold | $F_y$<br>$F_z$ | $(-10, 10)$ N<br>$(-10, 10)$ N |
| Track increment threshold | $\Delta_x$<br>$\Delta_y$<br>$\Delta_z$<br>$\Delta_\alpha$<br>$\Delta_\beta$<br>$\Delta_\gamma$ | 0 mm<br>1 mm<br>5/3 mm<br>0°<br>0.02°/3<br>0.02°/3 |
| Stiffness threshold | $d_x, d_y, d_z$ | $(0, 4000)$ |

Depending on the change of force in the direction of the $y$- and $z$-axis during assembly, the threshold of the contact force $F_y$ and $F_z$ is set to 10 N, which is set to 0° because the assembly in this article is independent of the displacement of the $x$-axis direction and the

rotation angle of the *z*-axis. The stiffness threshold is the maximum and minimum values that are desirable as the threshold.

### 5.2. Fuzzy Reward System

The contact force $F_y$ of the input t-moment *y* axis, the contact force $F_z$ of the *z*-axis, the assembly depth $Z$ of the t-moment axis, and the assembly action quantity $D_Z$ are fuzzed by the triangular membership function, whose respective membership functions are shown in Figure 6.

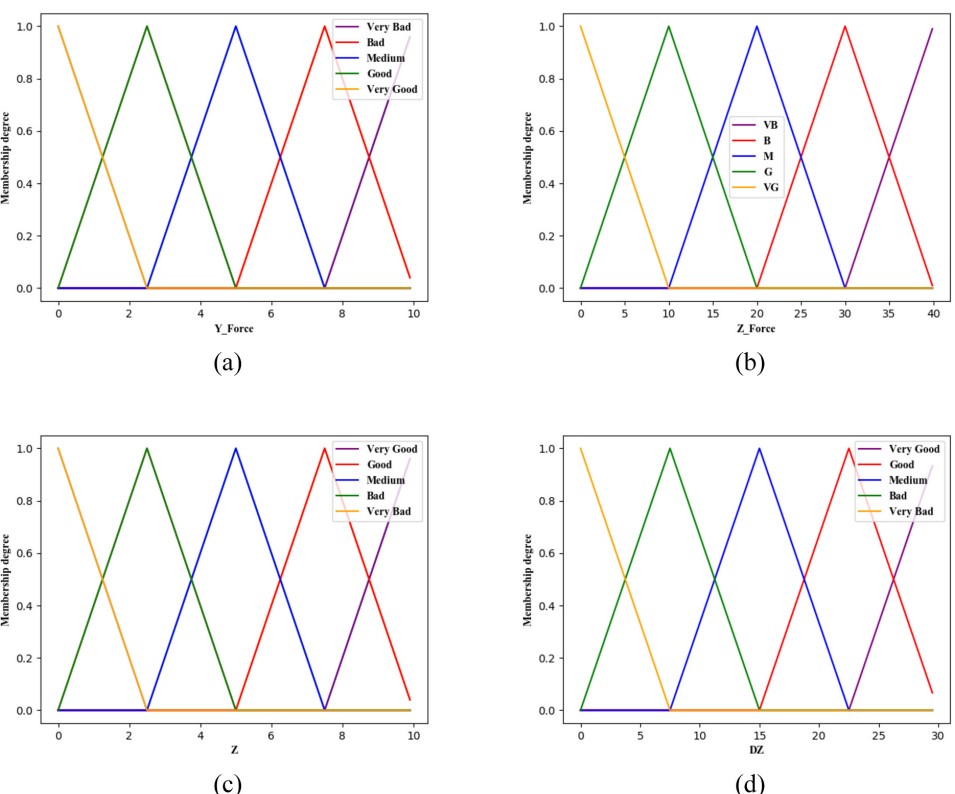

**Figure 6.** The first layer of membership function: (**a**) $F_y$ membership function; (**b**) $F_z$ membership function; (**c**) $Z$ membership function; (**d**) $D_Z$ membership function.

The fuzzy set after fuzzy is fuzzy according to the set rule library to obtain the output fuzzy set of the first layer; the output of the first layer is obtained by the deflating operation; and the output of the first layer is shown in Figure 7.

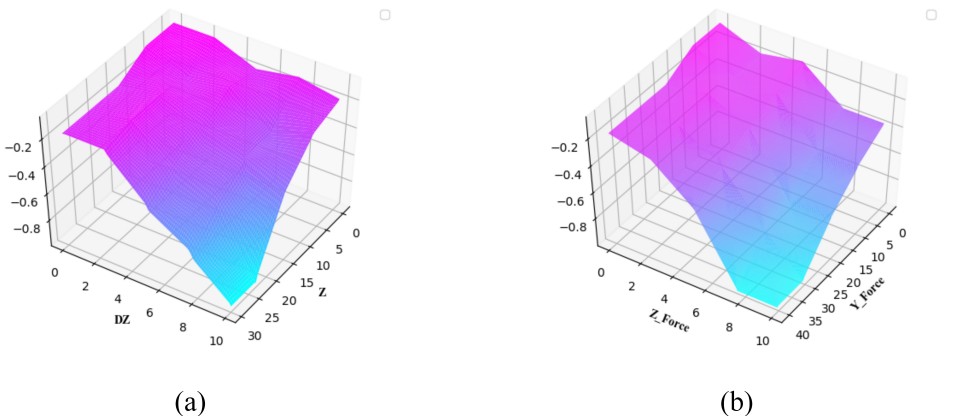

**Figure 7.** The first layer of fuzzy logic system output: (**a**) $F_y - F_z$ fuzzy logic system output value; (**b**) $Z - D_Z$ fuzzy logic system output value

The fuzzy logic output of the first layer is used as the input of the fuzzy logic system of the second layer, the same as the fuzzy logic system of the first layer, and the fuzzy operation is carried out first; its membership function is shown in Figure 8.

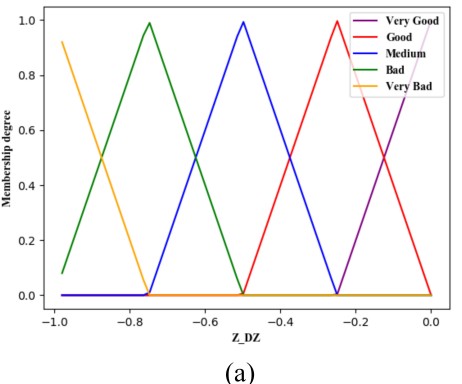

(a)

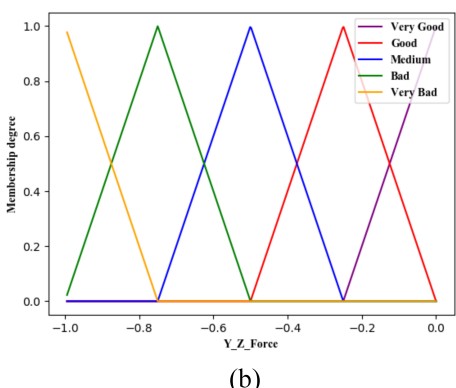

(b)

**Figure 8.** The second layer of membership function: (**a**) $F_y - F_z$ membership function; (**b**) $Z - D_Z$ membership function .

The resulting fuzzy set fuzzy reasoning, de-fuzzing and other operations to obtain the final evaluation quality are as shown in Figure 9.

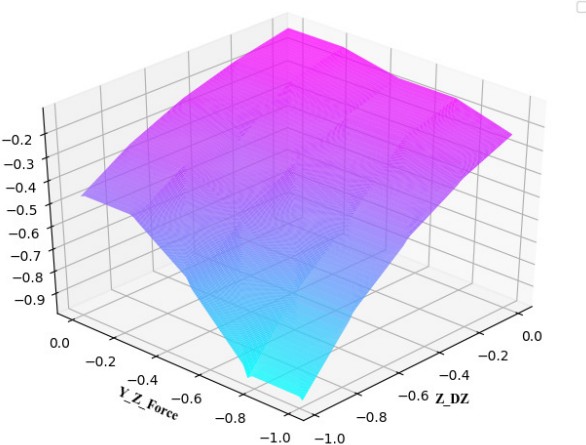

**Figure 9.** The second layer of fuzzy logic system output.

### 5.3. The Assembly Strategy Learning Process

The robustness and generalization capability of the model are improved, and the initial position introduces angle errors $(-0.05°, 0.05°)$ and positional errors $(-10\text{ mm}, 10\text{ mm})$ to simulate real industrial environments. In this experiment, the effectiveness of the algorithm is verified by comparing with the other two assembly methods. One is to use position control without impedance control to realize peg-in-hole assembly; the other is to use fuzzy logic to adjust impedance parameters. Other than that, the setup for each experiment is the same. The initial parameter settings for learning training are shown in Table 4.

**Table 4.** Training parameter setting of each experiment.

| Without Soft | | Soft with DDPG | | Soft with Fuzzy | |
|---|---|---|---|---|---|
| Symbol | Value | Symbol | Value | Symbol | Value |
| episode | 1000 | episode | 1000 | episode | 1000 |
| stepmax | 20 | stepmax | 20 | stepmax | 20 |
| $s_0$ | $s_0$ | $s_0$ | $s_0$ | $s_0$ | $s_0$ |

The training results of each experiment are shown in the Figure 10, including the loss value, reward value and step value during the learning process.

As can be seen from Figure 10, the loss value during training decreases and converges with the increase in step, indicating that the gap between the predicted action value of the Actor network output and the action value required by the actual environment is getting smaller and smaller.

The value of the reward in Figure 10 increases with the number of trainings, indicating that the learning is getting better and better, and the validity of the algorithm is verified.

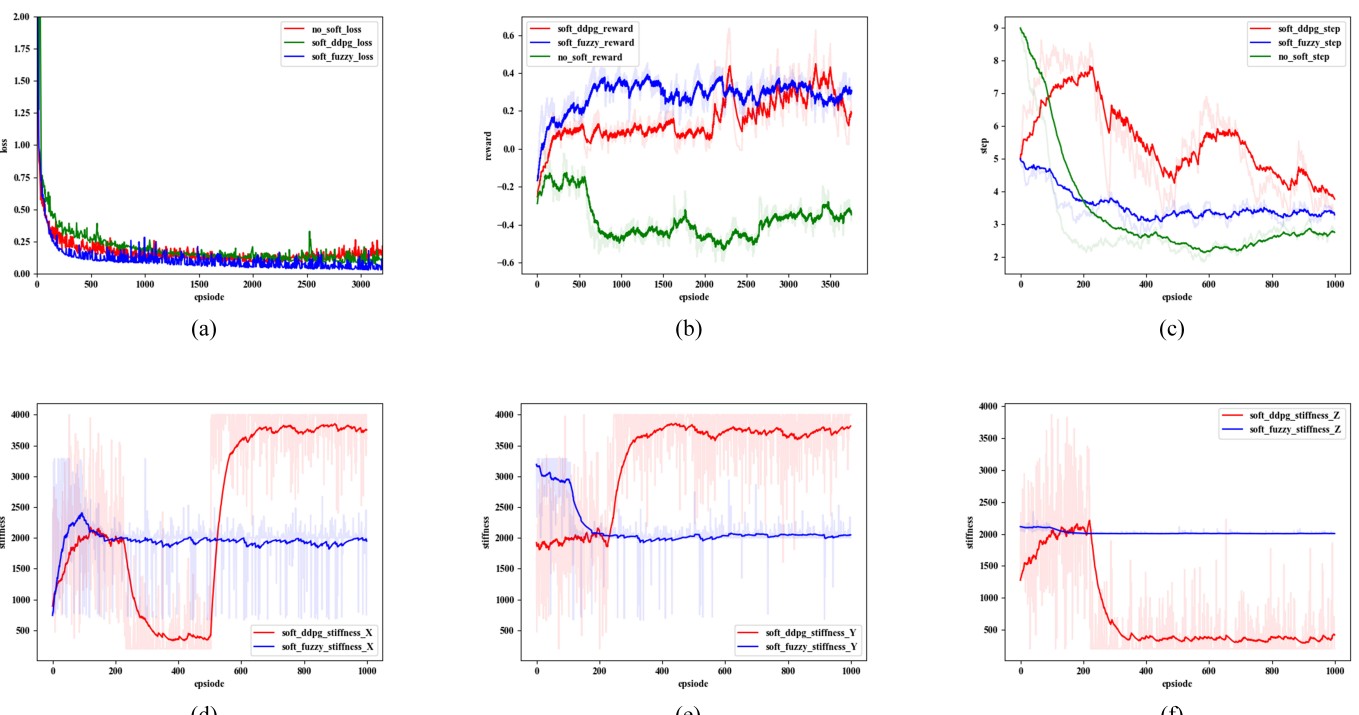

**Figure 10.** The assembly strategy learning process: (**a**) loss value change during training; (**b**) reward value change during training; (**c**) step value change during training; (**d**–**f**) represent the changes of the impedance values in the $x$, $y$, and $z$ directions when the impedance parameters are adjusted by the learning method and the fuzzy logic method, respectively.

The step value in Figure 10 requires six steps to complete the assembly task before the training begins, but only four steps after training to complete the assembly task, indicating that the assembly is getting faster and faster. Figure 10 shows the changes in impedance parameters during training.

The force variation curve of each experiment is shown in Figure 11. Without the addition of impedance control, the peg-in-hole assembly process generates a large contact force/moment. The magnitude of force/torque variation produced by the method of adjusting the impedance parameters using learning is smaller than that of adjusting the impedance parameters using fuzzy logic. The security and effectiveness of the algorithm are verified.

### 5.4. Assembly Strategy Experimental Validation

5.4.1. Test the Assembly Success Rate

After the training is completed, we save and test the trained model, a total of 10 tests, each time with 20 peg-in-hole assembly; the test results as shown in Table 5.

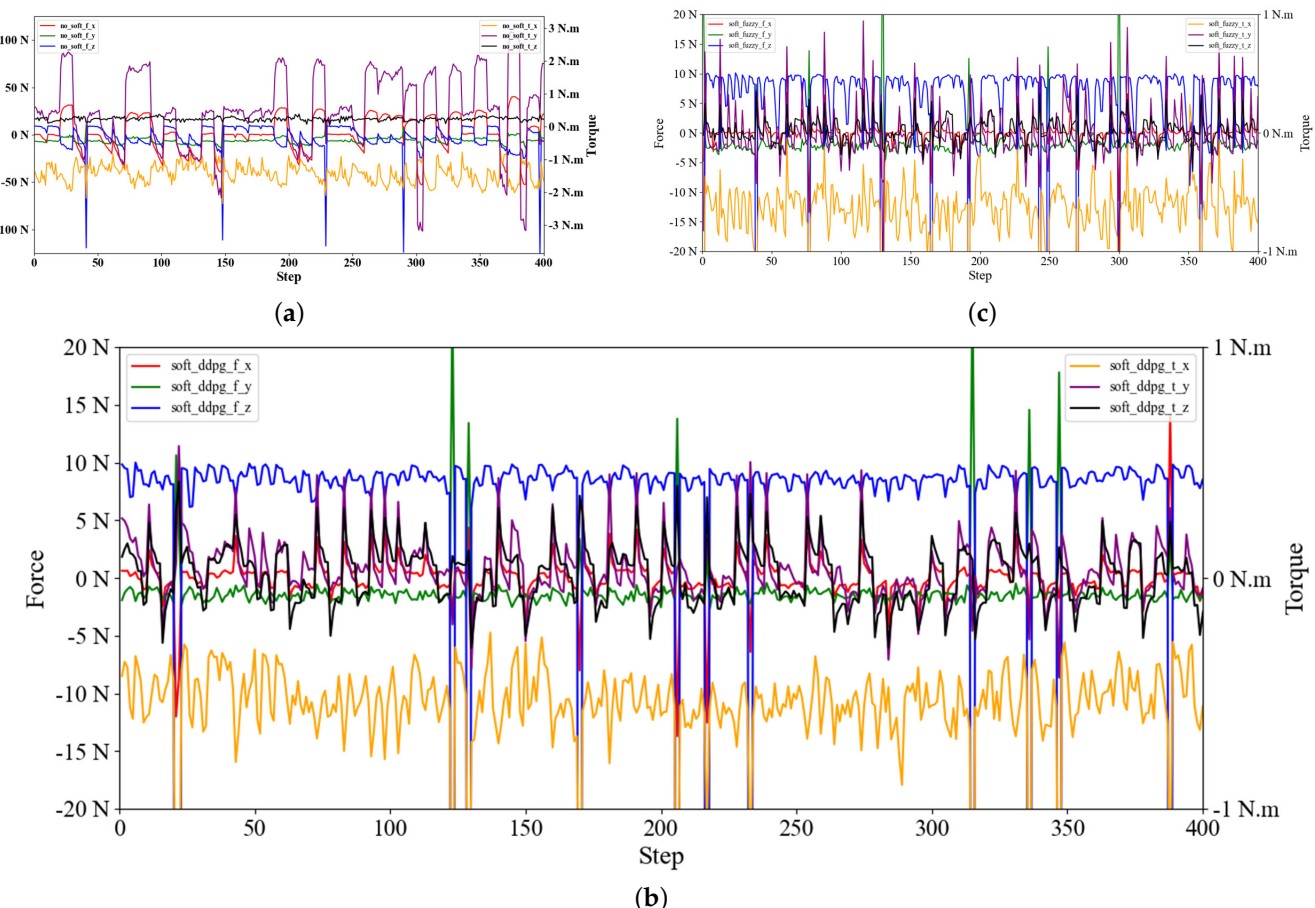

**Figure 11.** The force variation curve of each experiment. (**a**) Without soft; (**b**) soft with DDPG; (**c**) soft of fuzzy.

**Table 5.** Model test results.

| Groups | Without Soft Success | Soft DDPG Success | Soft Fuzzy Success |
|---|---|---|---|
| 1 | 0% | 95% | 85% |
| 2 | 0% | 100% | 85% |
| 3 | 0% | 95% | 80% |
| 4 | 0% | 100% | 90% |
| 5 | 0% | 100% | 80% |
| 6 | 0% | 100% | 85% |
| 7 | 0% | 100% | 75% |
| 8 | 10% | 100% | 65% |
| 9 | 0% | 100% | 80% |
| 10 | 0% | 90% | 90% |

Results of each model performance are showen in Figure 12. It can be seen from the test results that if there is no impedance control, the robot still cannot complete the peg-in-hole assembly task after training. The method of adjusting impedance parameters using fuzzy logic can achieve a success rate of more than 85%, which is less than the 100% success rate that can be achieved using learning to adjust impedance parameters.

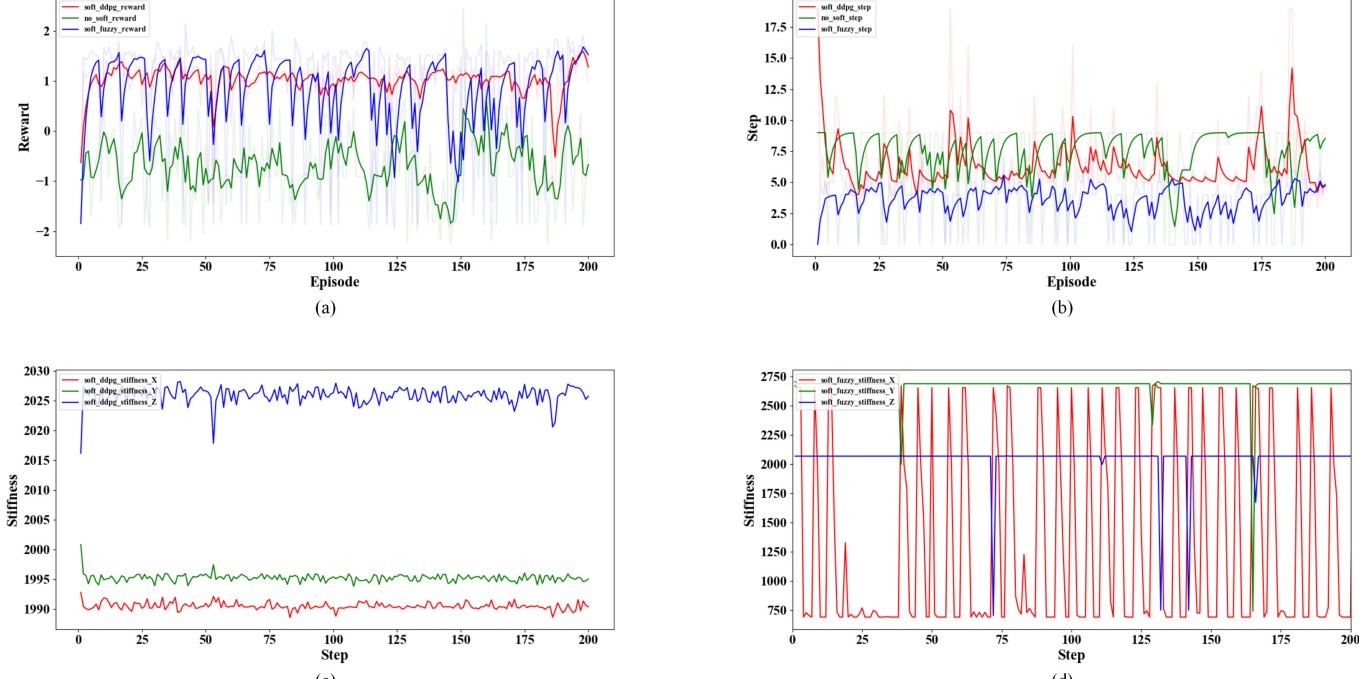

**Figure 12.** Results of each model performance: (**a**) reward; (**b**) step; (**c**,**d**): stiffness.

### 5.4.2. Assembly Process Analysis

The trajectory of a peg-in-hole assembly is sampled to obtain the y and z plane's motion trajectory as shown in Figure 13. The track is divided into the hole-in stage and the un-hole stage.

The joint torque in the assembly process, according to the sampled track point, reads the torque information of each track point in the entry and exit hole stage and the unaddressed stage respectively, and processes it, as shown in Figure 14. The torque of each joint does not change significantly in the unedited stage, and the mutation occurs in the hole-in stage, which gradually converges as the jack progresses. The larger changes are J1, J2, J3, and J4 joints.

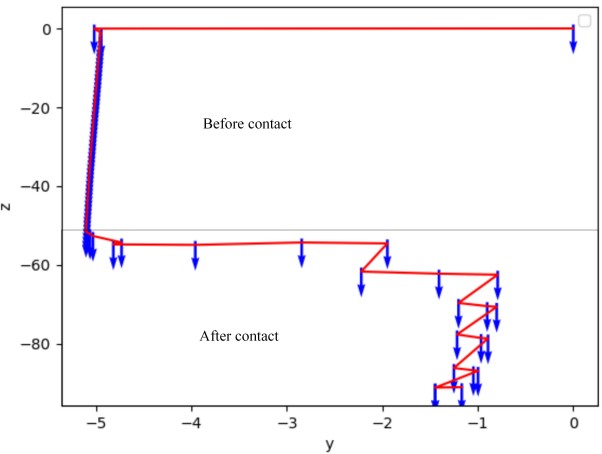

**Figure 13.** The trajectory of the end of the peg during insertion.

The end contact force in the learning process, according to the sampled track point, reads the end force information of each track point in the entry and exit hole stage and the unedited stage, respectively, including the *x*-axis end moment, the *y*-axis end force, the *z*-axis end force. The variation is shown in Figure 15.

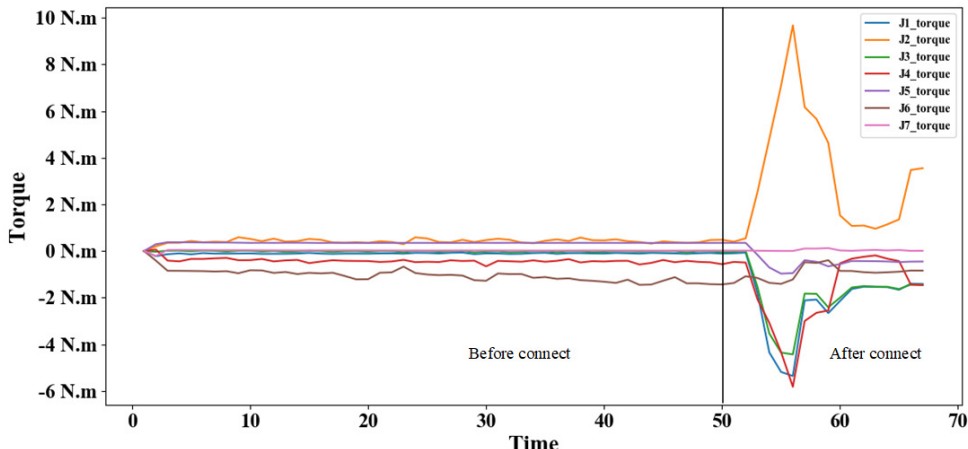

**Figure 14.** The torque of seven joints during assembly.

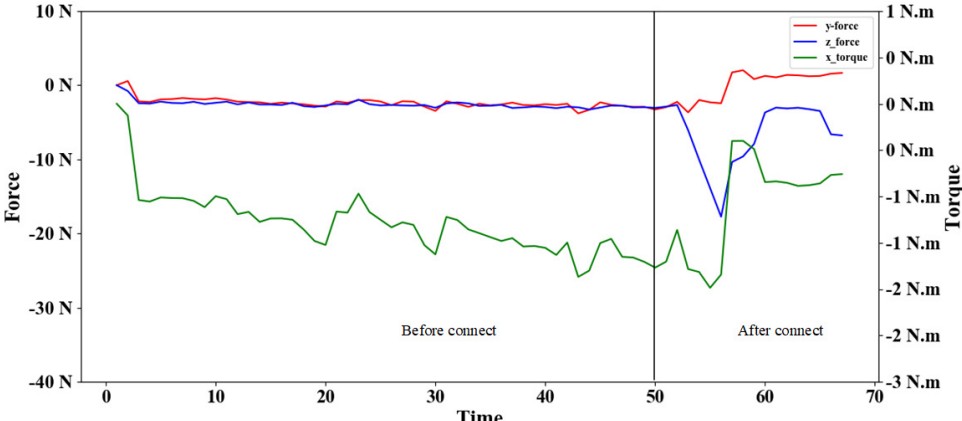

**Figure 15.** The end contact force and torque during assembly.

## 6. Conclusions

Robot automatic assembly of weak stiffness parts is a great challenge due to the interaction and high degree of uncertainty of contact force in the course of robot operation. This paper proposed a robot assembly skill learning system, combining the compliance control and deep reinforcement, which could acquire a better robot assembly strategy. The quality evaluation is designed based on fuzzy logic, and the assembly strategy is studied with the deep deterministic strategy gradient. The effectiveness and robustness of the proposed algorithm are demonstrated through a designed simplified simulation model and realistic peg-in-hole assembly experiments. However, the impedance control parameters of the robot are sensitive to the environmental position noise, and it is difficult to optimize the parameters in the industrial field environment. Since the design of the reward function is closely related to the assembly process, various factors must be considered to improve the learning efficiency. The future work will combine with model-based intensive learning to improve the learning efficiency and robot assembly performance.

**Author Contributions:** Z.W., Y.M. and F.L. wrote the reports; Z.W. and X.Y. read and organized the literature; R.S. and F.L. proposed verification schemes and experimental methods; F.L. and Y.M. carried out experimental platform construction and algorithm network construction; Y.M. and T.F. performed the compilation of experimental results. All authors have read and agreed to the published version of the manuscript.

**Funding:** This research was funded by Guangdong Key Research and Development Program (No.2020B090925001), by NSFC-Shenzhen Robot Basic Research Center project (U2013204), National Natural Science Foundation of China (No. 61973196).

**Institutional Review Board Statement:** Not applicable.

**Informed Consent Statement:** Not applicable.

**Data Availability Statement:** The study did not report any data.

**Conflicts of Interest:** The authors declare no conflict of interest.

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
