# Peer review of "Deep Deterministic Policy Gradient with Reward Function Based on Fuzzy Logic for Robotic Peg-in-Hole Assembly Tasks"

_applsci, doi:10.3390/app12063181_

Round 1

Reviewer 1 Report

Deep deterministic policy gradient with reward function based on fuzzy logic for robotic peg-in-hole assembly tasks

Purpose: Design and use a DRL method to define a policy for the robotic peg-in-hole task
Challenge: Architecture of the DRL networks, definition of the reward, matching of the selected DRL method with the peg-in-hole task
Application: Peg-in-hole task
Results: An efficient strategy is learned to perform the task, in simulation and on a real robot
Perspective: Improve learning efficiency using model-based learning
Main idea: Combining DRL and a reward based on fuzzy logic to achieve a given task

General plan : Introduction, Related Work, Method, Experiments, Conclusion

--

The paper and the proposed idea are timely and interesting.
The methodology is valid, and the results seems to be supported by the experiments.

However, the presented results are only sufficient to validate the feasibility of the approach. The paper lacks a comparison with the state-of-the-art approaches for the peg-in-hole task, in order to assess the potential benefits of this approach.

This makes it hard to understand the position of the paper relatively to the state-of-the-art.
Especially since some recent papers are addressing the same task similarly:
- Especially citation [18] using both DDPG and Fuzzy Logic for peg-in-hole task
- Also https://ieeexplore.ieee.org/abstract/document/9210190

Thus, it is required to highlight the contribution of the paper over the other SOTA methods. A comparison, either in simulation or on the real robot, would be ideal for this purpose.
Without this, it's hard to see the real novelty and the real contribution of the paper.

Proofreading is definitely required. Among others:
- all mentions of "pig-in-hole" must be replaced by the adequate term,
- some sentences are hard to understand, like "However, the contact force cannot accurately estimate the contact forces" in the conclusion,
- relu must be replaced by ReLU,
- there should always be a space after a comma or a period, a space before an opening parenthesis, etc.

Reviewer 2 Report

This is an interesting paper; however, there are several key issues that need to be addressed first.

The abstract could better clarify to-the-point some of the main contributions stemming from the experimental procedure.

Why does the numbering of sections starts at 0 (zero), with the Introduction? Also, when you refer to the sections (e.g., at the end of the introduction) you use Roman numbering instead of Arab – you need to be consistent with the formatting.

The contributions, listed at the end of the introduction, are specifically practical contributions to solve the problem. Could you also list a few theoretical contributions?

The problem formula in section 2 is based in which literature?

Likewise, the method could be supported on existing literature.

In short, my main concern regarding this paper is the lack of connecting the experimental setup and procedure undertaken with previous research works. This is clear from the last two sections: the experiments and testing seem robust, but there is no discussion to compare their results with previous work. This needs to be addressed, to clarify the contribution in terms of innovation to existing body of knowledge.

Round 2

Reviewer 1 Report

Most of my concerns have been addressed.

Line 87: kills must be changed to skills.

Reviewer 2 Report

The authors have concisely addressed my main concerns, thank you.